# Retinal Vessel Density in Age-Related Macular Degeneration Patients with Geographic Atrophy

**DOI:** 10.3390/jcm11061501

**Published:** 2022-03-09

**Authors:** Suji Hong, Mihyun Choi, Cheolmin Yun, Seong-Woo Kim

**Affiliations:** 1Department of Ophthalmology, Korea University Guro Hospital, College of Medicine, Korea University, 148, Gurodong-ro, Guro-gu, Seoul 08308, Korea; ghdghd3596@gmail.com (S.H.); ksw64723@korea.ac.kr (S.-W.K.); 2Department of Ophthalmology, Korea University Ansan Hospital, College of Medicine, Korea University, 123, Jeokgeµm-ro, Danwon-gu, Ansan 15355, Korea; yuncheolmin@korea.ac.kr

**Keywords:** geographic atrophy, optical coherence tomography angiography, age-related macular degeneration, intermediate AMD

## Abstract

We compared the retinal vessel density and inner retinal thickness in patients who had one eye with geographic atrophy (GA) and a fellow eye with intermediate age-related macular degeneration (iAMD). The vessel density from the superficial vascular complex (SVC) and deep vascular complex (DVC) through optical coherence tomography angiography and the thickness of the nerve fiber layer, ganglion cell–inner plexiform layer (GCIPL), inner nuclear layer (INL), outer nuclear layer (ONL) on a structural optical coherence tomography thickness map were measured in 28 eyes of 14 GA patients with iAMD in the fellow eye. GA eyes had significantly lower vessel density in the SVC (26.2 ± 3.9% vs. 28.3 ± 4.4%; *p* = 0.015) and DVC (24.2 ± 2.6% vs. 26.8 ± 1.9%; *p* = 0.003) than fellow eyes (iAMD). GCIPL and ONL were significantly thinner in GA eyes than in the fellow eyes (*p* = 0.032 and 0.024 in the foveal areas, *p* = 0.029 and 0.065 in the parafovea areas, respectively). Twenty-four eyes of 12 patients were followed up for 2 years and seven of the fellow eyes (58.3%) developed GA during the follow-up period and showed reduced vessel density in the SVC (26.4 ± 3.0% vs. 23.8 ± 2.9%; *p* = 0.087) and DVC (25.8 ± 2.2% vs. 22.4 ± 4.4%; *p* = 0.047) compared to baseline. Vessel density and GCIPL thickness map measurements are potential GA markers in non-neovascular AMD.

## 1. Introduction

Advanced age-related macular degeneration (AMD) is the leading cause of irreversible vision loss in developed countries [1]. While early stages of AMD present with drusen and pigmentary changes, advanced AMD includes geographic atrophy (GA) characterized by retinal pigment epithelium (RPE) and photoreceptor loss or the development of choroidal neovascularization (CNV) in neovascular (exudative) AMD [2]. RPE cell degeneration accompanied by the accumulation of drusen (lipo-glyco-proteinaceous deposits) is classified as early or intermediate AMD (iAMD) based on drusen size (i.e., small drusen <63 µm or intermediate druse *n* = 63–125 µm vs. large drusen ≥125 µm) [3].

GA has traditionally been defined on color fundus photographs as a discrete area of RPE atrophy measuring at least 175 µm in diameter observed together with photoreceptor loss [4]. Recently, the Classification of Atrophy Meetings (CAM) group termed GA as a subset of complete RPE and outer retinal atrophy without associated CNV: (1) a region of hyper-transmission of at least 250 mm in diameter, (2) a zone of attenuation or disruption of the RPE of at least 250 mm in diameter, and (3) evidence of overlying photoreceptor degeneration, in absence of signs of an RPE tear [5,6]. Although GA primarily affects the outer retina including photoreceptors and RPE, studies have previously demonstrated secondary inner nuclear layer and ganglion cell loss in addition to outer retinal atrophy [7,8,9]. Recently, reduced retinal vessel density in both the superficial and deep layer capillaries was reported in eyes with GA which is thought to result in reduced inner retinal metabolic demand in the GA area [10]. However, the reduced flow in the retinal capillaries was also often present in eyes with iAMD more than in age-matched controls [11,12]. To the best of our knowledge, there has been no study comparing the retinal blood flow in eye with GA and iAMD. Considering the AMD is a progressive retinal degenerative disease in which patients advance from the intermediate stages to GA, hypothetically there may be difference in the retinal blood flow between eye with GA and iAMD.

Optical coherence tomography (OCT) angiography (OCTA), which has recently been used clinically, has the advantages of being noninvasive, depth-resolved, and able to repeatedly measure the retinal capillary vessels [13]. OCTA scans can be obtained at the same location at different time points and can provide quantitative metrics of the retinal microvasculature of patients across consecutive visits [14]. However, those quantitative metrics from OCTA might be affected by systemic condition (including age, sex, hypertension, diabetes) and ocular biometry [13,15].

In this study, to minimize the confounding factors in analysis of retinal vessel density using OCTA, we compared the retinal vessel density and inner retinal thickness of both eyes in patients with one eye GA and fellow eyes with iAMD using OCTA. In addition, we sought correlations between retinal vessel density and size of GA.

## 2. Materials and Methods

### 2.1. Patient Selection

This study was conducted after obtaining approval from the Institutional Review Board of Guro Hospital, Korea University, and adhered to the tenets of the Declaration of Helsinki. The medical records of patients who were diagnosed with GA in one eye and iAMD in the fellow eye at Guro Hospital of Korea University between 1 March 2015 and 31 August 2021 were retrospectively analyzed.

The inclusion criteria included age >70 years and GA in the unilateral eye and iAMD in the fellow eye characterized by soft drusen larger than 125 µm within two disc diameters of the fovea and pigmentary changes. GA was defined following CAM report 3, as lesions of complete RPE and outer retinal atrophy that showed a region of hyper-transmission of at least 250 µm in length, a zone of attenuation or disruption of the RPE of at least 250 µm in length, overlying the ellipsoid zone degeneration on spectral-domain OCT imaging (Heidelberg Engineering, Heidelberg, Germany) without signs of RPE tear and CNV [5]. We included GA within the Early Treatment Diabetic Retinopathy Study (ETDRS) circle (6 mm) centered on the foveola to include the location of the GA in the OCTA scan. The iAMD classification was adopted based on the proposal by Ferris et al. in 2013 as follows [3]: patients with large drusen (>125 μm) or with pigmentary abnormalities associated with at least medium drusen (63–125 µm) should be considered to have iAMD.

Patients with neovascular AMD, small drusen measuring less than 63 µm, any history of anti–vascular endothelial growth factor (VEGF) injection, any past vitreoretinal surgery, or cataract surgery within six months of analysis; any maculopathy secondary to causes other than AMD (e.g., presence of diabetic or cystoid macular edema, epiretinal membrane, macular hole, or vitreomacular traction syndrome); and eyes with myopia greater than −3.0 D were excluded. Patients with ocular hypertension (intraocular pressure >21 mmHg), glaucoma, retinal degenerative disease (e.g., retinitis pigmentosa, cone-rod dystrophy, stargardt disease), or diabetic retinopathy were also excluded. All patients underwent a complete examination, which included best-corrected visual acuity (BCVA), slit-lamp examination, intraocular pressure, and fundus examination.

### 2.2. Image Acquisition and Analysis

Blue-light FAF imaging was performed using a confocal scanning laser ophthalmoscope (Spectralis HRA+OCT; Heidelberg Engineering, Heidelberg, Germany), with a scan angle of 30° × 30° (8.7 × 8.7 mm). Structural OCT and OCTA images were obtained using a spectral-domain OCT device (Spectralis OCT2; Heidelberg Engineering, Heidelberg, Germany). OCTA images were obtained as 4.3 × 4.3 mm (20° × 20°) angiography scans, using 384 B-scan images (high-speed mode), with a wavelength of 870 nm, lateral resolution of 11.64 μm/pixels, and axial resolution of 3.87 μm/pixels. OCTA images with artifacts, such as shadows, double-vessel patterns, or horizontal movement lines, were excluded from the analysis [16]. Low-quality OCTA scans that were out of focus or with media opacity, tilted scan with uneven OCTA signal, and abnormal fixation showing eccentric foveola on OCTA scans were also excluded.

The retinal vascular density was determined based on the projection-artifact removed OCTA images of the superficial vascular plexus (SVC), from the internal limiting membrane (ILM) to the inner surface of the inner plexiform layer (IPL), and the deep vascular plexus (DVC), from the outer surface of the IPL to the outer surface of the outer plexiform layer (OPL) [17]. In the case of segmentation errors, the boundary line of each B-scan image was manually adjusted by checking and refining the segmentation lines (ILM, IPL and OPL) in each structural OCT (Figure 1a). To avoid bias in manual segmentation, the OCTA flow signal was minimized to 0% and the readers adjusted the line by checking the structural OCT (Figure 1a). Vessel density was calculated as the ratio of pixels occupied by the vessels/all the pixels in the OCTA image [18] using ImageJ software (version 1.51; National Institutes of Health, Bethesda, MD, USA) after binarization using the Otsu auto-thresholding approach, which calculates optimum threshold by minimizing intraclass variances and maximizing interclass variances (Figure 1b,c) [19,20]. Although a gold standard for conducting binarization of retinal vessel density has not been proposed yet, Otsu’s method was suggested to have high repeatability in the binarization of retinal vessel density [21]. The central 0.6-mm diameter area centered on the fovea was excluded to mitigate the effect of the foveal avascular zone on vessel density measurements [22], as there is possibility of underestimating vessel density calculation in eyes with geographic atrophic eye, which did not affect the foveola.

An automated algorithm based on a directional graph search was used to segment the volumes in the HEYEX software program (Thickness map, Heidelberg Engineering, Heidelberg, Germany) [23]. The thickness of the inner retina and outer nuclear layer (ONL) was measured using a thickness map divided into nine areas of the ETDRS circle. In this study, the analysis included the central ring within 1 mm of the fovea, parafoveal ring (1–3 mm), and perifoveal ring (3–6 mm). Inner retinal layer thickness measurements were defined by the nerve fiber layer (NFL), from the ILM to the outer surface of the NFL, the ganglion cell–inner plexiform layer (GCIPL), from the inner surface of the ganglion cell layer (GCL) to the inner surface of the inner nuclear layer (INL), and by the INL. ONL thickness were measured from the outer surface of the OPL to external limiting membrane (ELM). Segmentations were reviewed in each structural OCT comprising the thickness map and manually adjusted by checking and refining the segmentation lines (ILM, NFL, GCL, INL, OPL and ELM) in each structural OCT to ensure their accuracy.

Measurement of GA size in the GA group. To assess the size of the GA on FAF images, the Regionfinder software (version 2.6.4.0; Heidelberg Engineering, Heidelberg, Germany), which automatically delineates atrophied lesions with hypo-autofluorescence (Figure 2), was used. Two ophthalmologists (S. H. and M. C.) measured the GA size, retinal vessel density, and retinal thickness independently of one another, and the averages of their reported values were used for analysis. To reduce the influence of baseline GA area on measured GA growth, square root transformation of GA area was implemented [24].

### 2.3. Follow-Up Analysis

Patients were followed up every three to six months. In patients with 24-month follow-up data (±3 months) from the baseline, we compared the retinal vessel density and inner retinal thickness in GA eyes at baseline and last follow-up (24-month) to elucidate GA-related changes. Of the 14 patients, 12 were able to participate in the 24-months follow-up. During the follow-up period, the development of GA in the fellow eye was also observed, and the retinal vessel density of fellow eye with iAMD at last follow-up was compared with that of the baseline.

### 2.4. Statistical Analyses

Data are presented as means ± standard deviation (SD). Prism 7 (GraphPad Software Inc., San Diego, CA, USA) was used for the statistical and graphical analyses. The Shapiro–Wilk test was used to test the normality of the data distribution of visual acuity, retinal layer thickness, size of GA, and vessel density of both eyes. As these values passed the normality test, an independent *t*-test was used to compare visual acuity, retinal thickness, and vessel density between each eye in an individual. A Pearson correction coefficient test was used to analyze the correlation between the size of the GA and retinal vessel density in the GA group. The paired t-test was used to analyze the progression of thickness and vessel density changes between baseline and follow-up in each eye. The Wilcoxon signed rank test was used in subgroup analysis that did not pass the normality test. All values were considered statistically significant at *p* < 0.05.

## 3. Results

Twenty-eight eyes of 14 patients with GA in one eye and iAMD in the fellow eye were included in the study. The study participants’ demographics are presented in Table 1. The mean age of study participates was 76.6 ± 5.1 and the mean follow-up period was 25.11 ± 5.93 months. The logarithm of the minimum angle of resolution (logMAR) BCVA of eyes with GA was 0.43 ± 0.50 at baseline and 0.70 ± 0.62 at last follow-up (*p* = 0.086), while that of the fellow eye was 0.30 ± 0.21 at baseline and 0.31 ± 0.27 at last follow-up (*p* = 0.752); thus, the BCVA was significantly worse in the eyes with GA at last follow-up (*p* = 0.050) (Table 1). The intraclass correlation coefficients of the two examiners for vessel density were 0.813 for SVC, 0.891 for DVC (both *p* < 0.001), and 0.855 for the measurement of GA size (*p* < 0.001).

### 3.1. Retinal Vessel Density and GA Growth

#### 3.1.1. GA Eyes vs. Fellow Eyes at Baseline

The retinal vessel densities in eyes with GA and fellow eyes at baseline were compared. The mean retinal vessel densities (%) were significantly lower in eyes with GA than in fellow eyes in the SVC (26.2 ± 3.9% vs. 28.3 ± 4.4%; *p* = 0.015) and DVC (24.2 ± 2.6 vs. 26.8 ± 1.9%; *p* = 0.003) (Figure 3).

#### 3.1.2. Follow-Up Analysis in Eyes with GA at Baseline

To assess the association between the change of the GA area and vessel density, the square root of the GA area and retinal vessel densities were compared between baseline and last follow-up. The square root of the GA area was 0.86 ± 0.55 mm (0.22–2.15 mm) at baseline in the GA group. In the twelve eyes that could be followed-up, the square root of the GA area was 0.92 ± 0.57 mm at baseline and 1.54 ± 1.03 mm at last follow-up, suggesting significant growth (*p* = 0.007) (Figure 4a). The retinal vascular density in eyes with GA at baseline that were available for 24-month follow-up analysis was 25.9 ± 4.5% in the SVC and 24.9 ± 7.5% in the DVC at baseline and 24.8 ± 2.9% in the SVC and 21.8 ± 4.9% in the DVC at last follow-up, which showed significant change only in DVC (*p* = 0.224 and *p* = 0.036) (Figure 4b). When analyzing the association between changes in the square root of the GA area, significant association was found between the change of square root of the GA area and vessel density of DVC. (*p* = 0.584 for SVC and *p* = 0.047, r = −0.583 for DVC).

#### 3.1.3. Follow-Up Analysis in Fellow Eyes

To assess the association between the development of GA and change of vessel density, retinal vessel densities at baseline and last follow-up of fellow eyes were evaluated. In the twelve eyes that were available for 24-month follow-up, the retinal vascular density in fellow eyes was 27.7 ± 4.0% in the SVC and 26.1 ± 2.1% in the DVC at baseline and 24.5 ± 4.1% in the SVC and 23.3 ± 4.2% in the DVC at last follow-up, which showed significant change in both layers (*p* = 0.011 and *p* = 0.024) (Figure 4c). Out of 12 eyes that were able to follow-up, seven fellow eyes (58.3%) newly developed GA during the follow-up period. The retinal vascular density in eyes which did not developed GA (*n* = 5) was 27.9 ± 4.8% in the SVC and 26.3 ± 2.6% in the DVC at baseline and 26.3 ± 4.8% in the SVC and 26.9 ± 2.1% in the DVC at last follow-up (*p* = 0.131 and *p* = 0.670) (Figure 4d). Of the eyes that developed GA (*n* = 7), retinal vessel density was 26.4 ± 3.0% at baseline and 23.8 ± 2.9% at last follow-up in SVC and 25.8 ± 2.2% at baseline and 22.4 ± 4.4% at last follow-up in the DVC, which showed a reduction and reached statistical significance in the DVC (*p* = 0.087 in SVC and 0.047 in DVC) (Figure 4e).

### 3.2. Inner Retinal Thickness (mm)

#### 3.2.1. GA Eyes vs. Fellow Eyes at Baseline

The mean inner retinal and ONL thickness in eyes with GA and fellow eyes at baseline were compared. The mean inner retinal thickness values measured in the central 1-mm circle (fovea), inner 1-mm to 3-mm circle (parafovea), and outer 3–6-mm circle (perifovea) of the ETDRS are presented in Table 2. The GCIPL thickness at the foveal and parafoveal areas was significantly thinner in eyes with GA than in fellow eyes (*p* = 0.032 and =0.029, respectively). The NFL thickness and INL thickness did not exhibit any significant differences between the eyes of the individual. The ONL thickness at the foveal and parafoveal areas was thinner in eyes with GA than in fellow eyes but reached statistical significance in the foveal area (*p* = 0.024 and =0.065, respectively).

#### 3.2.2. Follow-Up Analysis in Eyes with GA at Baseline

The values for the retinal thickness at baseline and at last follow-up (*n* = 12) in eyes with GA that were available for follow-up analysis were compared to assess the change of inner retinal thickness during follow-up period (Table 3). The GCIPL thickness was decreased in the outer 3- to 6-mm circle (perifovea) (*p* = 0.050), but the NFL and INL thickness did not show a significant reduction during the follow-up period. The ONL thickness was decreased in the outer ring 3- to 6-mm circle (perifovea) but did not reach statistical significance (*p* = 0.072).

## 4. Discussion

The results of this study showed that superficial and deep retinal vascular density in eyes with GA was significantly reduced compared to fellow eyes with iAMD. We included patients who showed unilateral GA and iAMD in their fellow eyes to minimize the co-factors that affect the retinal vessel density and inner retinal thickness. We further excluded patients with CNV or a history of anti-VEGF injection, so we can conclude that the lower retinal vessel density was not triggered by anti-VEGF therapy. Previously, You et al. compared the retinal vessel density of GA patients and age-matched control patients and reported that there was a reduction in vessel density from 9% to 13% in the former group [10]. Toto et al. reported that the vessel density of the superficial capillary plexus was decreased relative to that in healthy controls (48.7% vs. 50.4%), even in iAMD patients without outer retinal degeneration, while the reduction in SVC was not observed in patients with early AMD [11,12].

While the previous studies compared eyes with GA (or iAMD) and healthy control, we analyzed retinal vascular density in patients with unilateral GA and iAMD in their fellow eyes, and demonstrated the reduction of retinal vessel density in GA eye than that of fellow eye with iAMD. With photoreceptor atrophy in GA, the synaptic activity of the outer plexiform layer decreases, which is thought to be accompanied by a decrease in the vessel density of the DVC [10]. For the reduction in SVC, it was suggested that oxygen diffusion increases in the inner retina from choriocapillaris as a result of atrophic changes in the outer retina, which in turn induces relative constriction of the retinal vasculature, which is a similar mechanism as that involved in pan-retinal photocoagulation [25,26].

In this study, the GCIPL of GA eyes was significantly thinner in the center 1 mm and inner 1- to 3-mm ETDRS circles than in fellow eyes without GA and showed significant reduction at follow up analysis. Inner retinal deterioration in GA eyes has been well-demonstrated in histological and imaging studies. Kim et al. reported in a histological report of eyes with GA that the outer nuclear layer (ONL) was severely reduced (76.9%), while the GCL was also reduced by approximately 30%, and that INL cells were relatively preserved [9]. A volumetric and thickness analysis using OCT scans also reported a decrease in GCL, which was proportional to the decrease in ONL in GA patients [7,8]. It is assumed that a chronic decrease in stimulation from photoreceptors in patients with GA causes a secondary loss of ganglion cells.

A reduction in inner retinal thickness was also reported previously in eyes with iAMD, suggesting that the damage started in all retinal layers, not just the outer retina [12,27,28]. Borrelli et al. reported ganglion cell complex thinning in patients with intermediate AMD with OCT as well as reduced reflectivity in the inner and outer segments of photoreceptors and suggested photoreceptor neuronal loss in intermediate AMD [29]. Further, it is found that a 6% and a 2.5% longitudinal thinning of the GCL thickness in the fellow eyes with early and intermediate AMD of patients treated for neovascular AMD in one eye [30,31].

Although it is unclear whether the reduction in retinal vessel density in GA eyes is a secondary change due to photoreceptor loss or a concomitant preceding change before ONL atrophy occurs. The observation of a decrease in retinal vessel density and GCIPL thickness compared to fellow eyes with iAMD suggests several hypotheses. As the sequent neuro-retinal degeneration develops in intermediate stages AMD to GA, the neurodegenerative mechanism in iAMD might be followed by a reduction in oxygen demand and a resulting reduction in blood flow, or the inner retina might be damaged by progressive hypoperfusion secondary to vascular damage by AMD.

In the follow-up analysis, we observed growth in the size of the square root of GA area by approximately 0.62 mm during the 2 years. Previously, GA progression rates reported in the literature ranged from 0.53 to 2.6 mm^2^/year [32,33] and the presence of GA in one eye is a strong risk factor of future GA in the fellow eye [34]. There was a significant change in retinal vessel density in the deep vascular complex in the follow-up analysis of GA eyes and negative correlation between the change of square root of the GA area and vessel density of DVC. As the deep vascular complex is mainly located in INL, the vessel density reduction is thought to indicate a decrease in perfusion rather than the effect of volume reduction of INL, as there was no thickness reduction in INL at follow up, in comparison to baseline.

In the fellow eye analysis, the eyes that showed GA development during the follow-up period and reduction in retinal vessel density were found in the SVC and DVC, although statistical significance was confirmed only in DVC. We speculate that this result supports inner retinal deterioration, which, together with the loss of ganglion cell nuclei and axons, might be accompanied by the progression of GA with a loss of photoreceptors, followed by retinal capillary constriction. Retinal perfusion might be impacted by reduced metabolism due to neurodegenerative changes in the inner retina and reduced synaptic activity. The finding of the growth in GA size and changes in retinal vessel density of SVC during the follow-up period not having a significant correlation suggests that a reduction in SVC perfusion might be a secondary sequela rather than a cause of GA. Similarly, Seddon et al. reported that the loss of RPE in GA precedes the loss of choroidal vessels because the choroidal capillary is well-preserved at the edge of the lesion [35].

The limitations of the present study include its retrospective design and relatively small sample size. To overcome this limitation, we included patients who showed unilateral GA and iAMD in their fellow eyes, which might minimize the co-factors that affect the retinal vessel density and inner retinal thickness. A prospective study with a larger sample size is necessary to confirm our findings and to discern whether inner retinal changes precede retinal capillary changes. Furthermore, the square root of GA area in this study varied among patients from 0.22–2.15 mm, which might have produced an error that may have influenced the findings of retinal vessel density changes during follow-up, as both cases of early and advanced GA were included. In addition, during OCTA imaging in patients with advanced GA, segmentation errors might have occurred despite our efforts to eliminate such errors. Furthermore, we did not compare the flow voids in choriocapillaris between eyes with GA and iAMD, as we used spectral domain OCT with light at 870 nm wavelength, which was reported to have poor penetration into the choroid layer than swept source OCT (wavelength of 1050 nm) [36,37]; additionally, the drusen in iAMD eyes might induce signal attenuation of choriocapillaris layer [38]. Despite these limitations, to the best of our knowledge, this is the first comparison of vessel density of retinal capillary plexuses in patients with unilateral GA and could enrich our knowledge of the pathophysiology of GA.

## 5. Conclusions

In conclusion, the GA eyes showed a significantly reduced retinal capillary density in the SVC and DVC relative to fellow eyes with iAMD. In addition, there were significantly reduced thickness values of the GCIPL in GA eyes. This reduction in retinal vessel density is thought to be a secondary change to GA and might be a marker for GA secondary to non-neovascular AMD.

## Figures and Tables

**Figure 1 jcm-11-01501-f001:**
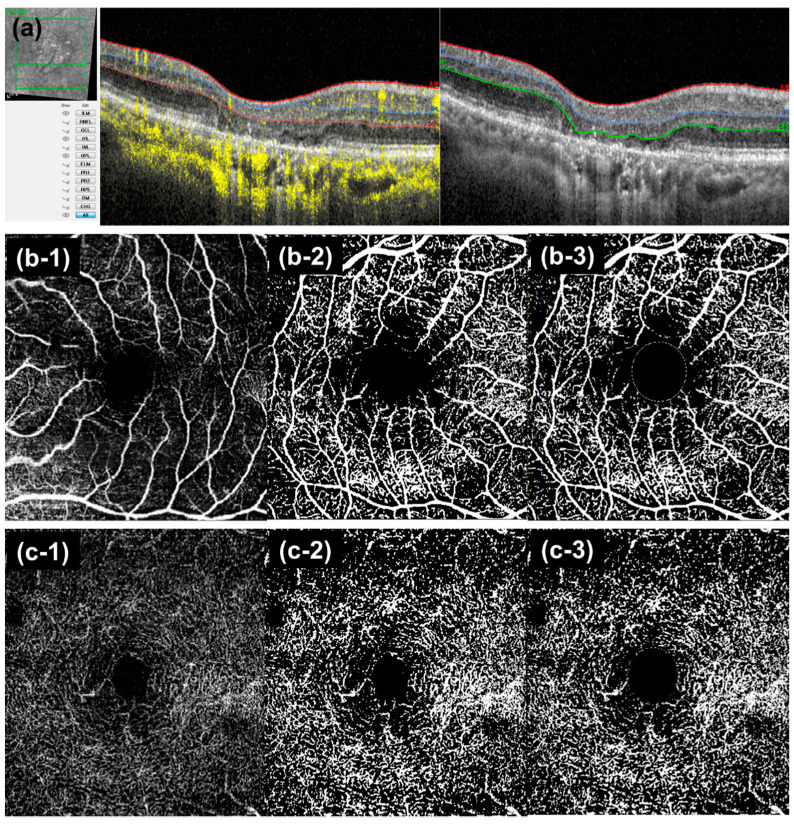
Optical coherence tomography angiography (OCTA) Image processing (**a**) (**Left**) Example of incorrect auto-segmented line in OCTA B-scan of eyes with GA. The segmentation line of inner plexiform layer and outer plexiform layer are misdrawn because of outer retinal atrophy. (**Right**) Manual adjustment of segmentation line. (**b**,**c**) Image processing in *En-face* OCTA image of the superficial vascular complex (SVC, (**b**)) and deep vascular complex (DVC, (**c**)). (**b-1**,**c-1**) Original image (**b-2**,**c-2**) of binarized image using Ostu’s auto-thresholding method. (**b-3**,**c-3**) In the binarized image, a central 0.6 mm sized circle was removed and the vessel density was calculated (pixels of vessel/[the pixel of total area −0.6 mm sized circle]).

**Figure 2 jcm-11-01501-f002:**
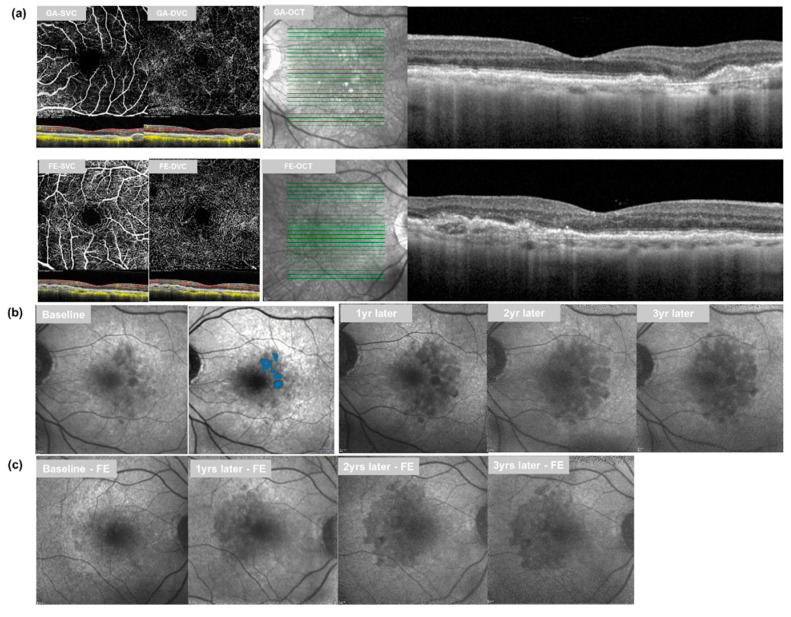
Multimodal images of patients in this study. (**a**) *En-face* optical coherence tomography angiography (OCTA) image of the superficial vascular complex (SVC) and deep vascular complex (DVC) with optical coherence tomography (OCT) + OCTA images (horizontal image at fovea) in eye with geographic atrophy (GA) (left upper) and fellow eye (FE) (left lower). On OCT + OCTA images, yellow dots show flow signal and red dotted lines represent the segmentation line for SVC and DVC slab. Horizontal spectral-domain OCT imaging in eyes with GA (right upper) and FE with iAMD (right lower). (**b**,**c**) Fundus autofluorescence (FAF) images of a case. A 76-year-old male patient who showed GA in the left eye and iAMD in right eye at baseline. (**b**) Measurement of GA lesions using Regionfinder is presented at baseline. The hypo-autofluorescence lesion on FAF showed significant growth during the follow-up period. (**c**) In the FE, a hypo-autofluorescence lesion was newly found in the FAF image 1 year later and a significant growth of GA was found 2 and 3 years later. GA: geographic atrophy; FE: fellow eye; SVC: superficial vascular complex; DVC: deep vascular complex.

**Figure 3 jcm-11-01501-f003:**
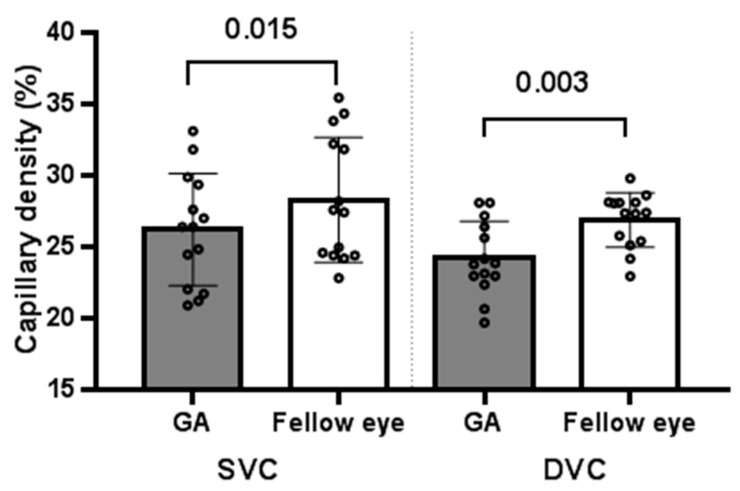
Mean vessel density of SVC and DVC in eyes with GA and fellow eyes. The mean retinal vessel densities were significantly lower in eyes with GA than the fellow eye. *p*-values are presented in the paired line above the bar, and error bar indicates the standard deviation. The position of each dot indicates values for an individual data point. GA: geographic atrophy; FE: fellow eye; SVC: superficial vascular complex; DVC: deep vascular complex.

**Figure 4 jcm-11-01501-f004:**
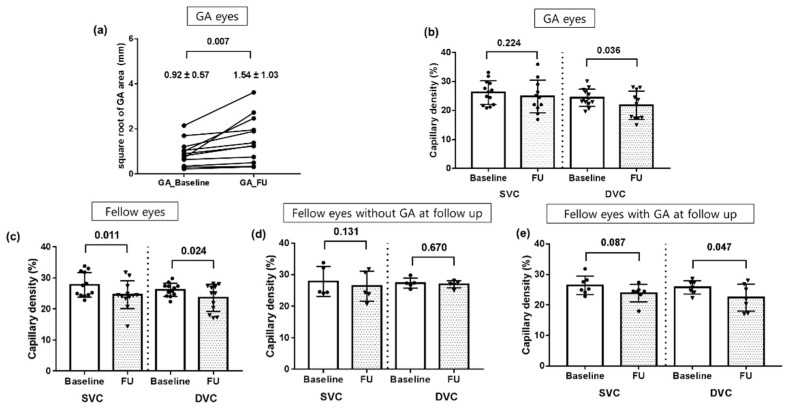
(**a**,**b**) Subgroup analysis in GA eyes with follow-up data available (*n* = 12). The size of the GA showed a significant growth during the follow-up period (*p* = 0.005), but the retinal vascular density at last follow-up showed significant difference in DVC (*p* = 0.036) but not in SVC (*p* = 0.224). (**c**) The retinal vessel density in fellow eyes at baseline and last follow-up (*n* = 12). The retinal vascular density at last follow-up showed significant reduction in SVC (*p* = 0.011) and DVC (*p* = 0.024). (**d**,**e**) Subgroup analysis of the retinal vascular density in eyes which did not developed GA (*n* = 5) and developed GA (*n* = 7) during the follow-up period. The reduction of retinal vessel density at last follow-up was found in eye which developed GA but reached statistical significance in DVC (*p* = 0.087 in SVC and 0.047 in DVC) *p*-values are presented in the paired line above the bar and the error bar indicates the standard deviation. The position of each dot indicates values for an individual data point. GA: geographic atrophy; FU: follow up; SVC: superficial vascular complex; DVC: deep vascular complex.

**Table 1 jcm-11-01501-t001:** Patient characteristics.

	Eyes with GA (*n* = 14)	Fellow Eyes (*n* = 14)	*p*-Value *
Age, years	76.6 ± 5.1		
Sex, male (%)	4 (28.6)		
HTN (%)	8 (57.1)		
DM (%)	4 (28.6)		
Eyes with GA, OD (%)	9 (64.3%)		
Follow up period (months)	25.11 ± 5.93		
Logmar BCVA (baseline)	0.43 ± 0.50	0.30 ± 0.21	0.466
LogMAR BCVA (last follow-up)	0.70 ± 0.62	0.31 ± 0.27	0.050
GA development in fellow eye during follow-up (*n* = 12) (%)		7 (58)	

Values are presented in mean ± standard deviation or with percentage. GA = geographic atrophy; HTN = hypertension; DM = diabetic mellitus; logMAR = logarithm of minimal angle of resolution; BCVA = best-corrected visual acuity. * Independent *t*-test.

**Table 2 jcm-11-01501-t002:** Comparison of inner retinal thickness values between both eyes.

		Eyes with GA (*n* = 14)	Fellow Eyes (*n* = 14)	*p*-Value *
Center ring 1 mm	NFL	12.14 ± 5.844	10.21 ± 2.26	0.246
	GCIPL	26.81 ± 6.82	34.00 ± 13.93	0.032
	INL	18.8 ± 7.29	17.9 ± 8.17	0.384
	ONL	63.29 ± 17.89	81.79 ± 22.77	0.024
Inner ring 1–3 mm	NFL	22.26 ± 3.61	21.71 ± 5.79	0.421
	GCIPL	75.68 ± 11.06	79.41 ± 11.03	0.029
	INL	37.02 ± 3.27	36.75 ± 4.64	0.919
	ONL	45.43 ± 7.02	51.96 ± 10.58	0.065
Outer ring 3–6 mm	NFL	36.62 ± 5.33	35.92 ± 9.75	0.310
	GCIPL	65.01 ± 6.65	64.36 ± 9.17	0.594
	INL	31.27 ± 1.81	30.55 ± 2.23	0.326
	ONL	43.77 ± 10.24	44.8 ± 9.74	0.786
Total (mean)	NFL	23.68 ± 3.92	22.63 ± 5.51	0.330
	GCIPL	55.84 ± 6.00	59.26 ± 9.17	0.033
	INL	29.03 ± 2.161	28.40 ± 2.95	0.285
	ONL	50.83 ± 6.42	59.52 ± 12.19	0.026

All values are presented as mean ± standard deviation. GA = geographic atrophy; NFL = nerve fiber layer; GCIPL = ganglion cell–inner plexiform layer; INL = inner nuclear layer; ONL = outer nuclear layer. * Independent *t*-test.

**Table 3 jcm-11-01501-t003:** Comparison of inner retinal thickness in eyes with GA between baseline and follow-up.

		Baseline (*n* = 12)	Follow-Up (*n* = 12)	*p*-Value *
Center ring 1 mm	NFL	13.33 ± 6.96	14.22 ± 5.95	0.891
	GCIPL	26.22 ± 6.74	24.88 ± 12.44	0.327
	INL	16.6 ± 5.45	18.44 ± 7.31	0.725
	ONL	62.6 ± 19.75	55.0 ± 18.44	0.162
Inner ring 1–3 mm	NFL	21.44 ± 4.05	35.53 ± 7.27	0.866
	GCIPL	75.47 ± 10.14	67.55 ± 18.91	0.233
	INL	35.93 ± 4.52	36.11 ± 4.69	0.779
	ONL	46.86 ± 12.76	43.28 ± 7.64	0.184
Outer ring 3–6 mm	NFL	35.25 ± 5.46	35.53 ± 7.27	0.779
	GCIPL	64.67 ± 7.39	57.61 ± 10.41	0.050
	INL	30.83 ± 2.67	30.25 ± 3.53	0.326
	ONL	46.7 ± 11.82	44.13 ± 12.42	0.072
Total (mean)	NFL	23.34 ± 4.80	23.76 ± 5.54	0.674
	GCIPL	55.45 ± 5.76	53.35 ± 10.41	0.594
	INL	27.81 ± 3.01	28.19 ± 4.15	0.889
	ONL	52.04 ± 11.63	47.47 ± 9.54	0.086

All values are presented as mean ± standard deviation. GA = geographic atrophy; NFL = nerve fiber layer; GCIPL = ganglion cell–inner plexiform layer; INL = inner nuclear layer; ONL = outer nuclear layer. * Paired *t*-test.

## Data Availability

The data that support the findings of this study are available from the corresponding author, Choi M, on reasonable request.

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
