# Peer review of "Retinal Vessel Density in Age-Related Macular Degeneration Patients with Geographic Atrophy"

_jcm, 2022, doi:10.3390/jcm11061501_

Round 1

Reviewer 1 Report

Hong et al. Retinal vessel density in non-neovascular age-related macular degeneration patients with unilateral geographic atrophy

Overall:

The manuscript addresses an interesting topic regarding OCT-a markers and superficial retinal thicknesses in patients with GA. The main conclusion is that GA coincides with lower vessel density and thinner inner retinal thicknesses than in iAMD.
This is an interesting message for eye researchers and clinicians. The reported results, however, warrant consideration. Descriptions and definitions of AMD phenotype are not up to date and somewhat confusing. Therefore, correct interpretation of the results is hampered. Overall, the writing and conclusions of the manuscript need careful attention, particularly in the results and discussion sections, see points below.

Title:
The definition of non-neovascular AMD may not be clear to readers. I would suggest to change the title to ‘Retinal vessel density in AMD patients with geographic atrophy”.

Abstract:
Line 12: ‘secondary to non-neovascular AMD’; not necessary to mention this. Geographic atrophy already implies there is no CNV and that it was preceded by iAMD, otherwise it is referred to as a mixed late phenotype in the literature. Change in entire manuscript.
Line 18: iAMD does not include presence of geographic atrophy according to most AMD classification systems; therefore change wording to GA patients with iAMD in the fellow eye.

Introduction:

Line 34: choroidal neovascularization (CNV) is the preferred nomenclature for neovascular AMD.

Line 34-35: ‘’in non-neovascular AMD’’; non-neovascular AMD implies GA. Please omit this part in the sentence, and in other parts of the manuscript.

Line 39: “the advanced form of non-neovascular” omit this piece. GA is already advanced AMD and not a complication of advanced AMD.
Line 41: This definition of GA is quite old. Currently, it is more standard to use and refer to the definitions of GA as proposed by the CAM report 4 (Guymer et al 2020 Ophthalmology).
Line 45: AMD is not widely considered a “neuro-”degenerative disease. It is considered a retinal degenerative disease, where mostly RPE and photoreceptors are involved. That being said, the authors studied the inner layers in the retina which indeed are involved in neurodegeneration. More explanation as to why these specific layers were investigated is needed in the introduction.
General: The gap of knowledge and corresponding hypothesis are not directly clear from this introduction. This should be highlighted. Why should we study retinal vessel density in AMD patients? The authors only describe results from studies, but their own hypothesis was not described.  

Methods:

Line 66-67: the definition of iAMD already excludes presence of CNV, omit “non-neovascular’’ before iAMD, and throughout the manuscript.

Line 78: The most recent consensus on GA was reported in CAM report 4 (Guymer et al 2020 Ophthalmology); authors should use this definition for GA inclusion.
Line 83: “Neovascular with exudative changes’: choose either, or ‘’patients with neovascular AMD’’ or ‘’patients with exudative changes’’

Line 83-88: How about glaucoma, or other neurodegenerative retinal diseases? Were such patients excluded? This is very important as inner retinal layers were examined.
Lines 137-146: How did authors deal with probable errors in automated segmentations? This needs to be described in more detail. Was the algorithm validated? For which circumstances did the algorithm fail? Were there any exclusions due to segmentation errors?
Lines 169: “(included + 3 months to – 3 months” Unclear what this means. Please elaborate. How many follow-up visits of what time interval were available for the patients?

Line 186-187: check the grammar; were versus was

General: Table 1 is included in the methods, but are referred to in the results.

Results

Table 1: One would expect the mean fellow eye at follow-up to be lower than at baseline, as some fellow-eyes developed GA; this needs to be discussed. Also, at what time point in the follow-up was the BCVA determined, at the last follow-up?

Line 212: “on the other hand ..” The use of On the other hand does not seem to make sense in this context.
Line 215: “and 21.8+-4.5 in the DVC..” add ‘’at follow-up’’ for clarification.

Line 218-219: “in 12 eyes that were able to..’ check grammar of sentence

3.1 general comment: the results section needs to be more structured; the results are now all summarized within one paragraph without clear context. Shortly introduce each analysis with a description of why it was done, and use different paragraphs. This will help readers to correctly interpret the results. Three main comparative analyses were done regarding mean vessel density: 1) GA vs fellow eye, 2) FU vs baseline, for eyes with GA at baseline 3) FU vs baseline, for eyes that developed GA at FU. This is not directly clear from the text.

Discussion:
General comment: Improve the overall structure of the discussion section. Particularly, the first section: lines 270-306. E.g. retinal vessel density results were compared using one paragraph, while retinal thickness results were compared using two separate paragraphs. I would also suggest to start with results from this study, rather than starting with results from other studies. This will help with the interpretability of the comparison.

Furthermore, the authors describe the results from other studies too elaborately and also mix in lengthy descriptions of biological mechanisms described by those. It would be better to first compare the results, thereafter discuss possible biological mechanisms, perhaps in a separate paragraph. Currently it is very difficult to find any context to the results from this particular study.

Reviewer 2 Report

The article by Hong and their colleague examines geographic atrophy (GA) in non-neovascular age-related macular degeneration patients. In particular, the authors focuses on the retinal vessel density and inner retinal thickness compared between eyes in the same patient with unilateral GA and investigated their potential relation to the development of GA. The authors show decrease in retinal vessel densities and inner retinal layers measured with OCT angiography in GA eyes compared with fellow eyes. They also reported that GA developed in over the half of fellow eyes during the follow up period with significant reduction in retinal vessel densities.

This paper will be of interest to researchers and clinicians involved in the basic research of non-neovascular age-related macular degeneration and its management for the prevention of GA.

Several problems on data analysis and presentation, as commented below, should be addressed.           

I have the following concerns:

1) Too many comparisons are made in the manuscript; in particular as shown in Table 2 and Table3. Considering a small sample size, I wonder if so many comparisons for fine-grained segmented macular area/layer has statistical meaning. P values should be somehow corrected for multiple testing.

2) The authors reported significant decrease of DVC vessel density in fellow eye which developed GA during the follow up period. They should also show, if available, changes in retinal vessel densities in overall fellow eyes, which may be presented like Figure 4(a), just as GA eyes. The data, if negative, will be useful information on the clinical course of the fellow eyes of GA for other researchers.

3) The first paragraph in Discussion should be revised to clarify the novelty of the present study and consistency with previous reports, though I completely agree with contents of the paragraph. Absence of a topic sentence makes it difficult to read out the authors’ main claim, which I guess is in the last two sentences. The authors should mention that selecting unilateral GA patients minimized confounding factors.

4) It is hard to grasp what is claimed in the fourth paragraph in Discussion. The authors should exclude remark on choriocapillaris in the first sentence as is irrelevant to what they investigated. Moreover, the connection between the first and second sentences is not clear. I guess the topic sentence of the paragraph is the third sentence, which should be moved to the beginning of the paragraph.

Reviewer 3 Report

I read the paper entitled Retinal Vessel Density in Non-neovascular Age–Related Macular Degeneration Patients with Unilateral Geographic Atrophy Special Issue: Clinical research Advances in Age-Related Macular Degeneration”. The topic of the article is interesting. The paper is very good structured and exact. The results could contribute to better knowledge of the pathophysiology of geographic atrophy.

Author Response

Thank you so much for your comment.